# Tracking Prostate Carcinogenesis over Time through Urine Proteome Profiling in an Animal Model: An Exploratory Approach

**DOI:** 10.3390/ijms23147560

**Published:** 2022-07-08

**Authors:** Alexandra Moreira-Pais, Rita Nogueira-Ferreira, Stephanie Reis, Susana Aveiro, António Barros, Tânia Melo, Bárbara Matos, José Alberto Duarte, Fernanda Seixas, Pedro Domingues, Francisco Amado, Margarida Fardilha, Paula A. Oliveira, Rita Ferreira, Rui Vitorino

**Affiliations:** 1LAQV-REQUIMTE, Department of Chemistry, University of Aveiro, 3810-193 Aveiro, Portugal; alexandrapais@ua.pt (A.M.-P.); stephanie.reis@ua.pt (S.R.); taniamelo@ua.pt (T.M.); p.domingues@ua.pt (P.D.); famado@ua.pt (F.A.); ritaferreira@ua.pt (R.F.); 2Laboratory for Integrative and Translational Research in Population Health (ITR), Research Center in Physical Activity, Health and Leisure (CIAFEL), Faculty of Sports, University of Porto (FADEUP), 4200-450 Porto, Portugal; jarduarte@fade.up.pt; 3Centre for Research and Technology of Agro Environmental and Biological Sciences (CITAB), Inov4Agro, University of Trás-Os-Montes and Alto Douro (UTAD), Quinta de Prados, 5000-801 Vila Real, Portugal; pamo@utad.pt; 4UnIC@RISE, Department of Surgery and Physiology, Faculty of Medicine of the University of Porto, Alameda Professor Hernâni Monteiro, 4200-319 Porto, Portugal; rmferreira@med.up.pt (R.N.-F.); asbarros@med.up.pt (A.B.); 5GreenCoLab-Green Ocean Association, University of Algarve, 8005-139 Faro, Portugal; s.aveiro@ua.pt; 6CESAM-Centre for Environmental and Marine Studies, Department of Chemistry, Santiago University Campus, University of Aveiro, 3810-193 Aveiro, Portugal; 7Institute of Biomedicine—iBiMED, Department of Medical Sciences, University of Aveiro, 3810-193 Aveiro, Portugal; barbaracostamatos@ua.pt (B.M.); mfardilha@ua.pt (M.F.); 8Cancer Biology and Epigenetics Group, IPO Porto Research Center (CI-IPOP), Portuguese Institute of Oncology of Porto (IPO Porto), 4200-072 Porto, Portugal; 9TOXRUN—Toxicology Research Unit, University Institute of Health Sciences, CESPU, CRL, Avenida Central de Gandra 1317, 4585-116 Gandra, Portugal; 10Animal and Veterinary Research Center (CECAV), University of Trás-os-Montes and Alto Douro (UTAD), 5000-801 Vila Real, Portugal; fernandaseixas@utad.pt

**Keywords:** aging, GeLC-MS/MS, urinary proteomics, prostate adenocarcinoma, retinol-binding protein 4, cadherin-2, enolase-1

## Abstract

Prostate cancer (PCa) is one of the most lethal diseases in men, which justifies the search for new diagnostic tools. The aim of the present study was to gain new insights into the progression of prostate carcinogenesis by analyzing the urine proteome. To this end, urine from healthy animals and animals with prostate adenocarcinoma was analyzed at two time points: 27 and 54 weeks. After 54 weeks, the incidence of pre-neoplastic and neoplastic lesions in the PCa animals was 100%. GeLC-MS/MS and subsequent bioinformatics analyses revealed several proteins involved in prostate carcinogenesis. Increased levels of retinol-binding protein 4 and decreased levels of cadherin-2 appear to be characteristic of early stages of the disease, whereas increased levels of enolase-1 and T-kininogen 2 and decreased levels of isocitrate dehydrogenase 2 describe more advanced stages. With increasing age, urinary levels of clusterin and corticosteroid-binding globulin increased and neprilysin levels decreased, all of which appear to play a role in prostate hyperplasia or carcinogenesis. The present exploratory analysis can be considered as a starting point for studies targeting specific human urine proteins for early detection of age-related maladaptive changes in the prostate that may lead to cancer.

## 1. Introduction

Prostate cancer (PCa) is the second most common cancer worldwide, with 1,414,259 new cases and 375,304 deaths in 2020, and its incidence continues to increase in many countries [1]. Currently, early detection and successful management of PCa are one of the most difficult and controversial issues in the scientific and medical communities [2]. There is prostate-specific antigen (PSA)-based screening for PCa, and men with high serum levels of PSA can undergo invasive transrectal ultrasound-guided biopsy (TRUS-biopsy), which is associated with the risk of infection and sepsis [3]. However, the PSA test is prone to false-positive and false-negative results, leading to unnecessary biopsies being performed in men without cancer [4]. Therefore, its clinical use as a standalone test remains controversial [3]. Indeed, 23% to 42% of all PCa cases detected in Europe and the United States may be the result of overdiagnosis [5]. These data highlight the urgent need for new approaches for PCa diagnosis. In this context, urine has been studied for the detection of PCa biomarkers [6]. Among the many positive aspects of using this biofluid are its ease and noninvasive collection and the relatively large quantities that are available in almost all patients [7]. The use of protein biomarkers in clinical trials is increasing, with proteomic approaches providing insights into the biological processes modulated by pathophysiological conditions [7]. In this way, proteomic approaches can aid in the diagnosis and management of diseases by providing information on disease onset and progression and even patients’ response and/or susceptibility to a particular treatment [8]. Thus, urine proteome profiling can be useful in the characterization of pathophysiological mechanisms, especially of prostate diseases due to the proximity between the organ and the body fluid [9,10]. Urine samples are indeed a rich source of informative proteins secreted/released by proximal organs such as the bladder and prostate, at concentrations exceeding those observed in blood-derived samples [11]. The differentiated epithelial cells of the prostate secrete proteins, namely PSA, prostatic acid phosphatase (PAP), prostaglandins, vimentin, and keratins, that can be detected in urine and may be involved in prostate carcinogenesis [9]. Therefore, analysis of urine proteome profiles during the progression of prostate lesions may provide insight into the biological processes underlying prostate remodeling. Proteins involved in these processes are expected to be biomarkers for different stages of carcinogenesis.

Animal models that mimic the natural history of human cancers are critical for a better understanding of the mechanisms associated with the development and progression of PCa, without being affected by lifestyle and comorbidities. The choice of an animal model is crucial for the successful translation of experimental data into clinical practice [12]. In the model of hormonally and chemically induced PCa, male rats exhibit tumors that are hormone-dependent and histologically similar to those in men [12,13], making them a good option for studying prostate carcinogenesis in its early stages.

The aim of our study was to gain new insights into the progression of prostate carcinogenesis through urine protein profiling. For this purpose, an animal model of chemically and hormonally induced adenocarcinoma was used. Two time points of disease were considered to reflect the progression of prostate lesions over time from preneoplastic to neoplastic changes. Urinary proteome profiling was based on GeLC-MS/MS, followed by bioinformatics analyzes, including association and correlation of target proteins with prostate histological scores. Zymography was also performed to analyze proteolytic activity in urine. Anthropometric data were also analyzed to investigate the development of cancer-related body wasting.

## 2. Results

Several biological variables were collected at necropsy (Table 1) to assess the effects of prostate carcinogenesis on body composition. Body weight decreased in both PCa1 (*p* < 0.01 vs. Cont1) and PCa2 (*p* < 0.05 vs. Cont2) rats, indicating the development of cancer-related body wasting. Both Cont2 and PCa2 animals showed an increase in body weight compared to Cont1 (*p* < 0.001) and PCa1 (*p* < 0.0001) animals, respectively. These results suggest an age-related effect, which can be verified by the body weight to tibial length ratio. In addition, PCa animals had higher prostate weight in both age groups (*p* < 0.0001, PCa1 vs. Cont1 and *p* < 0.0001, PCa2 vs. Cont2), suggesting prostate hyperplasia. The ratio of prostate to tibial length was also increased in both PCa groups (*p* < 0.0001, PCa1 vs. Cont1 and *p* < 0.0001, PCa2 vs. Cont2). These results suggest that the increase in prostate weight was not due to an increase in animal size as indicated by tibial length. Tibial length is an indicator of animal size. Since bone growth never stops in rats, normalization of absolute organ weight to tibial length is commonly performed in these animals [14]. The age effect on the ratio of prostate to tibial length was also evident (*p* < 0.0001, Cont2 vs. Cont1 and *p* < 0.0001, PCa2 vs. PCa1). The ratio of *gastrocnemius* to body weight showed that the older animals suffered a loss of muscle mass compared to their littermates (*p* < 0.001 Cont2 vs. Cont1 and *p* < 0.01 PCa2 vs. PCa1), which could indicate age-related sarcopenia. In contrast, adipose tissue appears to increase with age (*p* < 0.0001 Cont2 vs. Cont1 and *p* < 0.001 PCa2 vs. PCa1), with a decrease in the retroperitoneal mass depot and a simultaneous increase of the mesenteric mass one. The decrease in body weight observed in PCa2 animals (compared with Cont2) was due to a decrease in fat mass but not muscle mass, which may indicate an early stage of cancer cachexia.

The incidence of histopathological lesions of the prostate in each group is shown in Table 2, with a higher prevalence of neoplasia in the prostate of PCa2 rats. The coexistence of multiple lesions (dysplasia and neoplastic alterations) was observed in one animal of the PCa1 group (out of ten) and in ten (out of fourteen) animals of the PCa2 group. While the incidence of pre-neoplastic and neoplastic lesions was 50% in the PCa1 group, it was 100% in the PCa2 group. No prostatic lesions were observed in the animals of the Cont1 group, but littermates of the Cont2 group showed some dysplasia (four out of ten), and one animal had neoplastic changes in the prostate, which may be related to aging. No macroscopic and microscopic signs of bladder lesions were observed.

To identify prostate carcinogenesis-related proteins in urine, we performed a GeLC-MS/MS experiment for quantitative comparison between groups. Five animals from each group were randomly selected to this analysis (histological lesions and lesions scores are listed in Appendix A) and were individually analyzed (no pools were considered). The desalted urine samples were analyzed using SDS-PAGE, and the results showed a similar bands profile with little inter-individual biological variation (Appendix A). Whole gel lane analysis by LC-MS/MS retrieved 321 unique proteins with, at least, two unique peptides (Appendix A). Of all urinary proteins detected, only 219 were common to all the animals studied (five animals *per* group), as shown in the Venn diagram (Appendix A). Unsupervised PCA and supervised PLS-DA were used to visualize group separation based on proteome datasets by means of dimensionality reduction. PCA analysis showed no clustering of proteomic data *per* group (Figure 1A). PLS-DA showed a mild separation of the PCa2 group (Figure 1B). Aging appears to increase the heterogeneity of the urinary proteome, as noted for Cont2 and PCa2 groups (Figure 1A, B). To gain a better insight into the lesion profile, a pairwise PLS-DA was performed between consecutive lesion scores (Figure 1C,D). This analysis showed a slight separation of score 2 (corresponding to higher malignancy).

To find quantitative patterns between the proteins and the experimental groups or the histological prostate lesions, heatmap analysis was performed. When looking at experimental groups, two main clusters (PCa1 and Cont1 vs. PCa2 and Cont2) were highlighted, indicating a stronger effect of aging on the protein profile in urine, compared to PCa. Proteins involved in the acute-phase response are overrepresented in the urine of older animals (Cont2 and PCa2), whereas the biological process of renal system development is most represented in the urine of younger animals (Cont1 and PCa1; Figure 2A). When hierarchical comparative heatmap analysis was performed considering the score of histological prostatic lesions, two main clusters were observed (score 2 vs. scores 0 and 1), supporting the separation observed in Figure 1D. In the urine of animals with more severe prostatic lesions (score 2), proteins involved in fibrinolysis predominated, whereas in animals without or with less severe prostatic lesions (scores 0 or 1), the biological process of regulation of cell–cell adhesion mediated by integrin was more represented. In animals with scores 0 and 1, the most prevalent molecular function among urinary proteins was epidermal growth factor-activated receptor protein (Figure 2B).

Then, volcano plot analysis was performed to identify the proteins that were altered in urine by PCa or aging. These scatter plots show log2 of fold-change vs. −log10 of *p*-value and allow the identification of statistically significant changes in a very large proteome dataset (Appendix A). Considering fold-change variations with a *p* < 0.05, a total of 49 proteins were present in significantly distinct amounts between groups and are listed in Table 3. Curiously, no proteins were found with significant differences between prostate lesion scores. More detailed information (specifically the biological function and cellular component) on these 49 proteins and the others whose abundance did not reach statistical significance between groups can be found in Appendix A. We found that different proteins were modulated by the two time-points of prostate carcinogenesis: four proteins were up-regulated and two proteins were down-regulated in PCa1 compared to Cont1, whereas three and five different proteins were up-regulated and down-regulated, respectively, in PCa2 compared to Cont2. Most of these proteins are of cellular origin (according to String-db.org, accessed on 7 April 2022, v11.5). The age effect on the number of urinary proteins was also evident when comparing the Cont2 and Cont1 groups. However, most of the differentially expressed proteins appear to have been modulated by age in the PCa groups (33 proteins, PCa2 vs. PCa1). Most of them are involved in binding processes (mostly protein binding) and originate from the circulating immunoglobulin complex, the external side of the plasma membrane and extracellular space (according to String-db.org, accessed on 7 April 2022, v11.5), as expected in a body fluid. To consolidate these results, we compared them with recent human urinary proteomics data from the literature (Appendix A) [15,16,17]. With respect to PCa1 vs. Cont1, four of six proteins have already been identified in the urine of PCa patients, namely retinol-binding protein 4 (RBP4). In PCa2 vs. Cont2, four of eight proteins were observed in the urine of PCa patients, such as enolase 1 (ENO1). The common proteins between the present study and the clinical studies showed the same trend in terms of abundance.

According to the Human Protein Atlas (February 2022), several of these 49 proteins have already been detected in prostate tissue sections from patients with PCa. Interestingly, some of them have not been detected in healthy individuals, which could represent potential PCa biomarkers. For example, RBP4 (the major circulating carrier of retinol from the liver to peripheral tissues as the prostate) was not detected in the prostate of healthy subjects but was detected in the prostate of PCa patients (Table 3). Indeed, urinary RBP4 levels were elevated in PCa1 rats (compared with Cont1). Similar results were found in humans, where RBP4 was increased in the urine of PCa patients (Appendix A). Nevertheless, RBP4 does not seem to be specific for PCa, since its levels were higher in the serum of patients with PCa or other diseases [18,19]. However, it seems that age can decrease the levels of this protein (Cont2 vs. Cont1 and PCa2 vs. PCa1), suggesting that RBP4 may be a urinary biomarker for early stages of age-related prostate carcinogenesis. Analysis of the urinary proteomic profile by prostate histological score did not highlight this protein, suggesting that age and prostate carcinogenesis interact in modulating the proteome. Higher urinary levels of ENO1 appear to be associated with a more advanced stage of prostate carcinogenesis (PCa2 group). This glycolytic enzyme is thought to originate in glandular cells of the prostate, and elevated levels have already been detected in PCa tissue samples and in the urine of PCa patients (Appendix A). Lower urinary cadherin-1 levels were observed in advanced stages of prostate carcinogenesis (PCa2 vs. Cont2), and similar results have been observed in urine of PCa patients (Appendix A); however, high expression is expected in glandular cells of the prostate (Table 3). To better understand the role of proteins significantly expressed in PCa1 vs. Cont1 and PCa2 vs. Cont2 in prostate carcinogenesis, a correlation was established between their abundance in urine and the prostate histological score (Appendix A). RBP4 did not correlate with lesion score, as already evident in the heatmap (Figure 2B). ENO1 showed a positive correlation between its abundance and lesion score (r = 0.6374, *p* < 0.05), reinforcing the idea that ENO1 is associated with advanced disease stages. Cadherin-1 content correlated negatively with lesion score, confirming the MS data (PCa2 vs. Cont2, Table 3). Isocitrate dehydrogenase (IDH)2 content, a key enzyme in citrate metabolism, correlated negatively with the prostate lesion score (r = −0.6696, *p* < 0.05), suggesting impairment of oxidative bioenergetics during the progression of prostate carcinogenesis. Interestingly, epidermal growth factor (EGF), which is associated with PCa development [20], showed a negative correlation with prostate lesion score (r = −0.6366, *p* < 0.05), indicating that it is not involved in more advanced stages of the disease.

Cachexia is associated with an overall catabolic state of the organism, which is reflected in increased proteolytic activity of the urine [21]. To evaluate this proteolytic activity in the context of prostate carcinogenesis, we performed a zymography analysis of urine samples. This analysis showed a similar proteolytic profile between groups in terms of the number of bands and the corresponding molecular weight (MW). In fact, five bands were observed to show activity (Figure 3A). The bands corresponding to the enzymes MMP9 (band 1; approximately 82 kDa) and MMP2 (band 2; approximately 66 kDa) showed no difference in activity between groups. The same trend was observed for bands 3 (approximately 30 kDa) and 4 (approximately 25 kDa), which appear to correspond to kallikrein-related peptidase (KLK)5 (MW: approximately 32 kDa; according to Uniprot) and KLK10 (MW: approximately 29 kDa; according to Uniprot), respectively (Figure 3B). Immunoblot analysis of MMP2 and MMP9 confirmed these results after finding no differences in the amount of MMP2 and MMP9 in urine between groups (Figure 3C). These results suggest that prostate carcinogenesis and aging do not affect urinary proteolytic activity. Furthermore, the zymography data do not support an overall catabolic status associated with PCa or aging, as observed in other cancers [21].

## 3. Discussion

Our study highlights the impact of the progression of prostate carcinogenesis on urinary protein profile, which allows putative biomarkers of early disease with screening and/or diagnostic value to be found. A large-scale GeLC-MS/MS strategy was used for proteomic profiling of urine from rats with chemically and hormonally induced prostate adenocarcinoma. This is a well-established animal model for PCa, in which tumors of the dorsolateral and anterior prostate develop, and was implemented as previously described [22]. The dorsolateral and anterior prostate of rats correspond to the areas of the human prostate that are more predispose to cancer development [23]. Histological examination of the prostate confirmed the greater neoplasia susceptibility of PCa2 animals (histological score 2), whereas PCa1 animals had only non-neoplastic or pre-neoplastic lesions (histological scores 0 and1). These results are comparable to the usual slow development of PCa in humans, which is preceded by dysplastic lesions [24]. Indeed, the incidence of PCa increases with age with approximately 25% of men diagnosed with PCa being 75 years or older [25].

Analysis of the urinary proteome showed that some proteins are altered by the progression of prostate carcinogenesis. The lower urinary levels of cadherin-1 or epithelial (E)-cadherin (PCa2 vs. Cont2) may be related to the down-regulation of this protein in cancer cells. This may be due to loss of heterozygosity, mutations or transcriptional silencing, a common event in PCa. Failure of cadherin-1 activity leads to loss of cell–cell adhesion, which promotes metastasis formation [26]. In fact, lower plasma levels of cadherin-1 have already been reported in PCa patients compared to the control group [27].

At this stage of carcinogenesis, no changes in the urinary levels of the other identified cadherin, cadherin-2 (also known as neural (N)-cadherin), were detected, in contrast to what was observed at the earlier time point (PCa1 group). Abnormal prostate expression of N-cadherin results in increased circulating levels of this mesenchymal marker [28]. Moreover, N-cadherin levels inversely correlated with E-cadherin content, and the dysregulation of both proteins seems to be involved in PCa development [29]. However, in contrast to E-cadherin, N-cadherin does not appear to play a role in the stages of prostate carcinogenesis studied.

Elevated levels of T-kininogen 2 were detected in the urine of PCa2 rats. It was previously reported that kininogen 1, the human homolog, is elevated in the urine of men diagnosed with PCa [21]. This protein can activate the mitogen-activated protein kinase (MAPK) pathway and promote the proliferation and survival of PCa cells [30]. EGF is also involved in the regulation of prostate growth and function, and its expression is increased in early and localized PCa; however, in our study, we observed a decrease in its levels in the urine of PCa2 animals and in animals with score 2 lesions, which may be explained by the stage of prostate carcinogenesis [31,32]. However, decreased EGF expression was found in elderly patients with benign prostate hyperplasia [33].

The high RBP4 levels detected in the urine of PCa1 animals may indicate increased retinol uptake by prostatic epithelial cells. Retinol carried by RBP4 is taken up by extrahepatic cells and converted to retinoic acid (RA). RA migrates to the nucleus, where it binds to nuclear receptors and regulates the expression of genes involved in the growth of epithelial cells in the prostate [34]. In particular, high levels of all-trans RA have been reported to slow prostate tumor cell proliferation and induce apoptosis [35]. This seems to be particularly relevant when testosterone metabolism is impaired. Indeed, RA has been reported to inhibit and reverse testosterone-induced lesions [36]. Thus, RBP4 content may reflect an attempt by epithelial cells to counteract prostate carcinogenesis when testosterone acts as a promotor (as is the case in this study). The increase in RBP4 levels has already been detected in the urine of patients with PCa [15]. RBP4 levels decreased with age, likely indicating age-related hormonal changes that may have contributed to the lack of a significant correlation between RBP4 levels and prostate lesion score.

Increased urinary ENO1 concentrations in the PCa2 group suggest increased expression of this glycolytic protein in tumor cells. Indeed, ENO1 correlated positively with prostate histological score. Moreover, increased urinary ENO1 levels have already been observed in PCa patients [16]. ENO1 is thought to energetically feed tumor development and progression by supporting anaerobic glycolysis; however, its pro-tumorigenic role goes beyond energetic purposes. At the cell surface, ENO1 acts as a plasminogen receptor and is involved in angiogenesis, tumor cell invasion and migration. Moreover, its secretion is stimulated by estradiol [37], whose circulating levels were significantly higher in PCa2 rats (data not shown). Furthermore, our results support the involvement of ENO1 in prostate aging, which may explain the dysplasia observed in prostate sections of some Cont2 animals. Aging is a recognized player in the development of benign prostatic hyperplasia and is associated with age-related hormonal changes. The increase in fat mass (observed in older animals compared with younger counterparts) may be responsible for an increased expression of aromatase and, consequently, estrogen production. In this way, free estradiol remains constant and affects the balance between estrogen and testosterone [38]. Thus, the age-related increase in ENO1 levels may indicate the metabolic reprogramming of prostate cells, which may lead to pre-neoplasia and, eventually, neoplasia.

In the PCa2 group, lower urinary levels of IDH2, a key enzyme in the Krebs cycle, were detected, and a negative correlation with the prostate lesion score was found. The abundance of this enzyme seems to oscillate during the cell cycle in an opposite manner to glycolysis [39]. Indeed, opposite changes in ENO1 and IDH2 levels were observed in PCa subjects. However, increased IDH2 levels were observed in aged healthy animals, which, together with no changes in ENO1 content, suggests that the aged healthy prostate relies more on glucose oxidation rather than glycolysis (as in PCa2 prostates) for energy production.

Changes in the urinary content of some other proteins appear to reflect the age-related remodeling of the prostate, with clusterin (Clu) as an example. Pronounced expression of Clu in dysplastic lesions has been reported and associated with prostate cells survival [40]. Neprilysin has previously been associated with PCa. It has been shown that down-regulation of the expression of this protein is due to extensive hypermethylation of the promoter region of the gene [41]. However, its transcription is up-regulated by androgens [41], which may explain why no changes were observed in PCa groups, but only in Cont2 compared to Cont1. Thus, decreased neprilysin levels may be considered a risk factor for the development of PCa. Corticosteroid-binding globulin (Serpina6) was detected in prostate tissue from patients with benign prostate hyperplasia [42]. The higher abundance of this protein in the urine of older rats (Cont2 vs. Cont1) may be responsible, at least in part, for the benign lesions observed in the prostates of these animals.

Overall, urinary proteomic profiling appears to provide important clues as to the time course of prostate carcinogenesis. Because of its proximity to the prostate, urine is the sample of choice for studying prostate remodeling, as highlighted here. Moreover, the use of an animal model allowed the study of the impact of PCa severity and aging on the urinary proteome profile without being biased by external factors such as lifestyle. Indeed, the data from this study support the analysis of urinary proteins to follow the time course of prostate carcinogenesis. Elevated levels of RBP4 and cadherin-2 signal early stages of prostate carcinogenesis, while ENO1, T-kininogen 2 and IDH2 represent more advanced stages. Urinary proteomic profiling also underscores the aging effect in prostate remodeling, which is characterized by elevated levels of Clu and Serpina6. The aging effect should be further explored in future urinary proteomic studies to identify protein markers of prostate aging that may predict the risk of PCa development.

## 4. Materials and Methods

### 4.1. Animal Protocol

The following protocol was approved by the Responsible Organ of animal well-being from University of Trás-os-Montes and Alto Douro and by the Portuguese Ethics Committee for Animal Experimentation (Direção Geral de Alimentação e Veterinária, license no. 021326). Forty-two, four-week-old male Wistar Unilever rats were acquired from Charles River Laboratories (France). They were randomly housed 5–6 in a cage for two weeks with *ad libitum* food and water access. The room temperature and the relative humidity were controlled at 18 ± 2 °C and 55 ± 5%, respectively, with a 12-h light-dark cycle.

Prostate lesions were induced as previously described [43]. In brief, twenty-four rats (PCa groups) were submitted to a subcutaneous injection of 20 mg·kg^−1^ of flutamide (prepared in 10% of propylene glycol and 5% of ethanol) during twenty-one consecutive days. After two days, these animals were subcutaneously injected with 100 mg·kg^−1^ of testosterone propionate (dissolved in starch oil). Then, after two days, the rats were intraperitoneally injected with 30 mg·kg^−1^ of the carcinogen *N*-methyl-*N*-nitrosourea (MNU; prepared in 0.1 M of citrate buffer, pH 4.8). Fifteen days later and under anesthesia (75 mg·kg^−1^ of ketamine and xylazine), subcutaneous implants with crystalline testosterone were placed in the rats’ interscapular region through a small incision followed by suture. The implants were prepared with medical silicone tubes, of which 4 cm were filled with testosterone, and the extremities were sealed with medical glue (G.E. RTV-108).

The animals were sacrificed at two different time points (counting from the induction of cancer): (i) at the twenty-seventh week, eight animals from the control group (Cont1) and ten animals from the PCa group (PCa1); and (ii) at the fifty-fourth week, the remaining ten animals from the control group (Cont2) and fourteen animals from the PCa group (PCa2). They were sacrificed by a ketamine and xylazine overdose, followed by cardiac puncture and exsanguination. The prostates were collected and processed for light microscopy by a routine histological approach. The *gastrocnemius* muscle and the adipose tissue (retroperitoneal and mesenteric) were also collected and weighted.

### 4.2. Histological Analysis

The prostate was fixed in 4% paraformaldehyde and embedded in paraffin. The resulting paraffin blocks were sectioned using a manual microtome. Prostate sections were deparaffinized in xylol, rehydrated with alcohol in decreasing concentrations (100%, 95% and 75%) and stained with haematoxylin and eosin (H&E). Lesions were classified blindly into groups according to Bosland [44]. For the association of urine proteome profile with prostate histological lesions, the following scores were considered: 0, no lesions; 1, dysplasia, prostatic intraepithelial neoplasia (PIN) or carcinoma; 2, coexistence of multiple lesions.

### 4.3. Urine Collection and Preparation

Urine samples were collected one week before each sacrifice. For that, rats were individually placed in metabolic cages for 12 h. Urine was centrifuged at 1000 g for 10 min at 4 °C and the supernatant was concentrated using 10 kDa filters (Vivaspin^®^ 500–10 kDa, Sartorius Biotech, Goettingen, Germany). The final retentate was resuspended in 100 µL of 0.5 M Tris, pH 6.8, and 4% SDS. Total protein content was estimated in the fraction corresponding to the retentate using the RC DC™ assay kit (Bio-Rad^TM^, Hercules, CA, USA) following the manufacturers’ instructions and using bovine serum albumin as standard.

### 4.4. SDS-PAGE and In-Gel Digestion

To a certain volume of urine samples corresponding to 50 µg of total protein (from five animals *per* group) was added in a proportion of 1:2 (*v*/*v*) the sample loading buffer (0.1 M Tris-HCl pH 6.8, 4% (*w*/*v*) SDS, 15% (*v*/*v*) glycerol, 20% (*v*/*v*) β-mercaptoethanol, 0.1% (*w*/*v*) bromophenol blue). The urine proteome was then separated in a 12.5% SDS-PAGE gel (37.5:1 acrylamide/bisacrylamide) resolved at 180 V in a Tris-glycine-SDS running buffer (25 mM Tris, 192 mM glycine and 0.1% SDS (*w*/*v*)). The gel was stained with 0.4% (*w*/*v*) Coomassie Blue G250 prepared in 50% of ethanol and 10% of acetic acid. Following the SDS-PAGE, complete lanes were cut out of the gel and sliced into seven sections. Each section was *in-gel* digested with trypsin. The resulting peptide mixture was then extracted from the gel fractions, dried using vacuum centrifugation and stored at −80 °C for posterior analysis.

### 4.5. LC-MS/MS Analysis and Protein ID

The dried extracted peptides were dissolved in 30 μL of mobile phase A (2% acetonitrile and 0.1% formic acid). Tryptic digests were then separated using an Dionex UltiMate 3000 RSLC UHPLC system (Thermo Electron Corporation, Waltham, MA, USA), coupled to a Q-Exactive hybrid quadrupole-Orbitrap mass spectrometer (Thermo Electron Corporation, USA) and equipped with an Easy-spray ion source and a PepMap RSLC C18 EASY-Spray column (2 μm particle size, 100 Å, 75 μm × 15 cm, Thermo Fisher Scientific, Waltham, MA, USA).

The resulting DDA data were used for protein identification and quantification using MaxQuant software (Max Planch Institute of Biochemistry, Planegg, Germany, v 1.5.6.5). Raw MS data were searched against the UniProt *Rattus norvegicus* protein database (downloaded in February 2018), and only curated proteins and canonical sequences were considered. A precursor mass tolerance of 20 ppm, a fragment mass tolerance of 0.1 Da and two missed cleavages were allowed for the search. Variable modifications were determined to be oxidation (M) and acetylation (N term of the protein), and fixed modifications were set for carbamidomethylation of cysteine. A protein level of 1% false discovery rate (FDR) was set to filter the result. Only proteins identified by at least 2 peptides were considered. Normalization of the data between LC-MS runs was performed in MaxQuant based on the summation of total peptide ion signals *per* sample and using a global Levenberg–Marquardt optimization procedure that aimed to minimize the overall changes for all peptides across all samples. Label-free quantification (LFQ) was then performed using MaxQuant based on the determination of the mean pairwise common peptide ratios between samples, requiring at least two peptide ratios *per* identified protein. The MS proteomics data have been deposited to the ProteomeXchange Consortium via the Proteomics Identifications (PRIDE) partner repository [45] with the dataset identifier PXD034432.

The biological process(es) to which each protein belongs was annotated in Appendix A based on the Perseus software platform (Max Planch Institute of Biochemistry, Planegg, Germany, v1.6.15.0). Jvenn [46] was used to identify proteins common to all groups. String ([47] v11.5) was used to identify biological processes, molecular functions or cellular components that include proteins modulated by PCa or aging. Before the following analyses, the raw data were log-transformed. Principal component analysis (PCA) and partial least squares discriminant analysis (PLS-DA) were applied to examine proteome dynamics between groups. Heatmap was used to examine the clustering of samples from each group or each prostate histological score according to protein abundance. Volcano plots were used to annotate proteins that were present in significantly different amounts between the groups or between prostate histological score. These analyses were performed using the MetaboAnalyst v5.0 web portal (www.metaboanalyst.ca, accessed on 10 March 2022). Four separate independent *t*-tests were carried out using LFQ data to compare Cont2 vs. Cont1, PCa1 vs. Cont1, PCa2 vs. Cont2 and PCa2 vs. PCa1.

### 4.6. Gelatin Zymography Analysis

Zymography assay was performed according to Caseiro and collaborators [48]. Briefly, 20 µg of urinary protein of each concentrated urine sample were incubated with charging buffer (0.1 M Tris-HCl pH 6.8, 5% (*w*/*v*) SDS, 20% (*v*/*v*) glycerol and 0.1% (*w*/*v*) bromophenol blue) for 10 min at room temperature in a proportion of 1:2 (*v*/*v*) and separated in a 10% SDS-PAGE gel with 0.1% (*w*/*v*) of gelatin. After the run, the gels were incubated in renaturation buffer (2.5% Triton X-100) for 30 min at room temperature with soft agitation, followed by incubation in the development buffer (50 mM Tris pH 7.4, 5 mM NaCl, 10 mM CaCl_2_, 1 µM ZnCl_2_ and 0.02% Triton X-100) for 30 min at room temperature with soft agitation. The gels were changed into a new development buffer and incubated overnight at 37 °C and then stained with 0.4% (*w*/*v*) Coomassie Blue G250 prepared in 50% of ethanol and 10% of acetic acid. Lastly, the gels were de-stained with a solution of 25% of ethanol and 5% of acetic acid. The gels were scanned with the ChemiDoc™ XRS+ system (Bio-Rad™, Hercules, CA, USA) and analyzed with Image Lab™ software (Bio-Rad™, Hercules, CA, USA, v6.0.0).

### 4.7. Immunoblot Analysis

Urine analysis by immunoblot was performed according to Ferreira and colleagues [21]. Briefly, concentrated urine samples of each animal were diluted in Tris-buffered saline (TBS) to a final protein concentration of 0.1 µg·µL^−1^. A volume of 100 µL was slot-blotted onto a nitrocellulose membrane (Hybond^®^ ECL™, Amersham Pharmacia Biotech, Amersham, Buckinghamshire, UK). Nonspecific binding was blocked with 5% (*w*/*v*) nonfat dry milk in TBS with Tween 20 (TBST). Each membrane was then incubated with the appropriate primary antibody solution [anti-matrix metalloproteinase (MMP)9 (ab38898) diluted 1:500 and anti-MMP2 (ab37150) diluted 1:200 from Abcam (Cambridge, UK)]. Membranes were then washed, incubated with the secondary antibody diluted 1:10,000 [IRDye^®^ 800 CW Goat anti-rabbit IgG (926-32211) from LI-COR Biosciences (Lincoln, NE, USA)] and washed again. All antibody solutions were diluted in 5% (*w*/*v*) nonfat dry milk in TBST and all incubations were performed for 1 h at room temperature with agitation. Detection was performed with fluorescence according to the manufacturers’ instructions (LI-COR Biosciences, Lincoln, NE, USA). Images were acquired using LI-COR Odyssey^®^ Scanner (LI-COR Biosciences, Lincoln, NE, USA) and analyzed using Image Studio™ Lite software (LI-COR Biosciences, Lincoln, NE, USA, v5.2.5). Optical densities (OD) values were expressed in arbitrary units.

### 4.8. Statistical Analysis

All analyses reported here were performed with biological replicates (no technical replicates were performed). Values are present as mean ± standard deviation for each experimental group or as percentages, where applicable. The Shapiro–Wilk test was applied to check the normality of the sample distribution. Significant differences (anthropometric parameters, LFQ data, zymography and immunoblot analyses) were determined by One-way analysis of variance (ANOVA) followed by Tukey’s multiple comparisons post-hoc test. Person correlation test was also performed to correlate the proteins whose abundance was statistically significant between PCa1 and Cont1 and PCa2 and Cont2 with the histological score of the prostate. The results were considered significantly different when *p* < 0.05. All these analyses were performed using GraphPad Prism^®^ software for Windows (GraphPad Software, San Diego, CA, USA, v6.0).

## Figures and Tables

**Figure 1 ijms-23-07560-f001:**
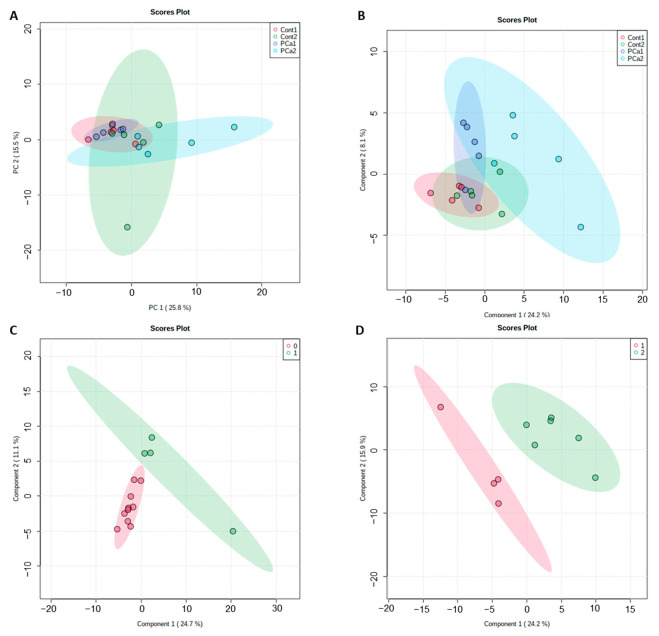
(**A**) Principal component analysis (PCA) and (**B**) partial least squares discriminant analysis (PLS-DA) of urine proteome data from animal groups (Cont1, PCa1, Cont2 and PCa2; five animals *per* group). Pairwise PLS-DA of urine proteome data among sequential prostate histological score (**C**) 0 and 1 and (**D**) 1 and 2.

**Figure 2 ijms-23-07560-f002:**
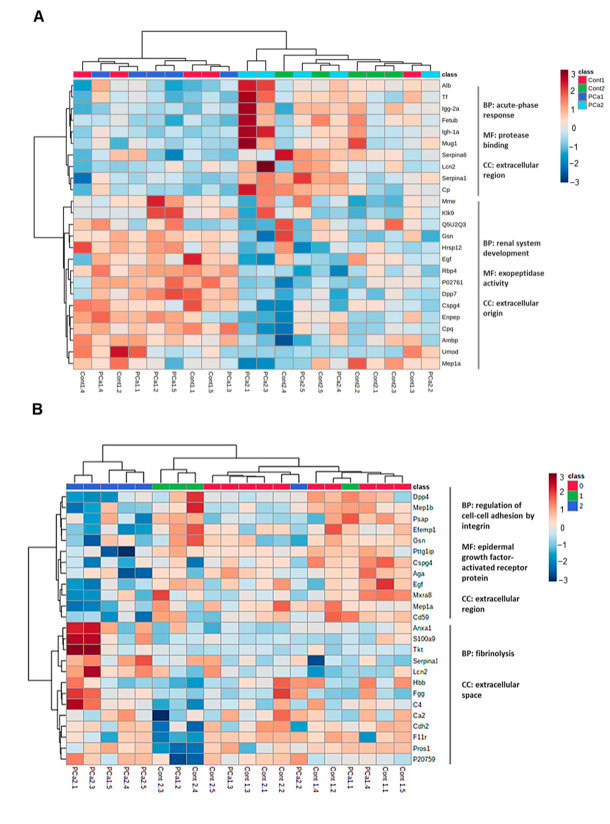
Heatmap representing the clustering of (**A**) animal groups showing two main nodes related to the biological processes: acute-phase response and renal system development and of (**B**) prostate lesion score highlighting two main nodes related to the biological processes: regulation of cell–cell adhesion mediated by integrin and fibrinolysis. The color from blue to red represents low to high protein abundance. BP, biological process; MF, molecular function; CC, cellular component (according to String-db.org, accessed on 7 April 2022, v11.5).

**Figure 3 ijms-23-07560-f003:**
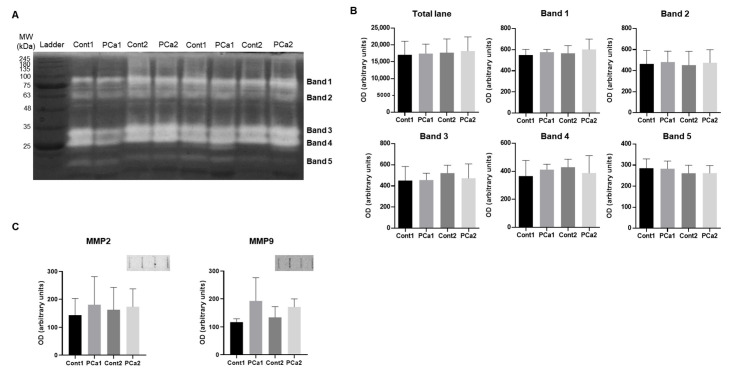
(**A**) Representative zymography gel of urine samples from different groups. (**B**) Optical density (OD) measurements of proteolytic bands. (**C**) Immunoblot analyses of urine levels of MMP2 and MMP9.

**Table 1 ijms-23-07560-t001:** Effect of prostate cancer on anthropometric data. Values are expressed as mean ± standard deviation.

	Cont1 (*n* = 8)	PCa1 (*n* = 10)	Cont2 (*n* = 10)	PCa2 (*n* = 14)
Body weight (g)	471.7 ± 27.8	408.5 ± 23.3 **	541.8 ± 44.9 ***	494.6 ± 35.4 ^####,+^
Body weight-to-tibial length ratio (g·cm^−1^)	53.0 ± 3.3	46.8 ± 2.5	119.4 ± 9.5 ****	112.5 ± 8.1 ^####^
Prostate weight (g)	2.7 ± 0.3	4.4 ± 0.4 ****	2.9 ± 0.3	4.8 ± 0.7 ^++++^
Prostate-to- tibial length ratio (g·cm^−1^)	0.3 ± 0.1	0.5 ± 0.1 ****	0.6 ± 0.1 ****	1.0 ± 0.1 ^####^^,^^++++^
*Gastrocnemius* muscle weight (mg)	4.7 ± 0.2	4.1 ± 0.4 **	4.4 ± 0.3	4.5 ± 0.3
*Gastrocnemius*-to-body weight ratio (mg·g^−1^)	1.0 ± 0.1	1.0 ± 0.1	0.8 ± 0.1 ***	0.9 ± 0.1 ^##^
Adipose tissue weight (g)	3.8 ± 0.6	3.4 ± 0.4	12.8 ± 3.5 ****	9.8 ± 1.9 ^###,+^
Adipose tissue-to-body weight ratio (mg·g^−1^)	0.8 ± 0.2	0.8 ± 0.1	2.3 ± 0.5 ****	2.0 ± 0.4 ^####^
Retroperitoneal adipose tissue weight (g)	2.8 ± 0.4	2.7 ± 0.3	0.9 ± 0.4 ****	0.8 ± 0.4 ^####^
Retroperitoneal adipose tissue-to-body weight ratio (mg·g^−1^)	0.6 ± 0.1	0.6 ± 0.1	0.2 ± 0.1 ****	0.1 ± 0.1 ^####^
Mesenteric adipose tissue weight (g)	1.0 ± 0.2	0.7 ± 0.1	11.8 ± 3.3 ****	9.1 ± 1.8 ^####,+^
Mesenteric adipose tissue-to-body weight ratio (mg·g^−1^)	0.2 ± 0.1	0.2 ± 0.1	2.2 ± 0.5 ****	1.8 ± 0.3 ^####^

** *p* < 0.01 vs. Cont1, *** *p* < 0.001 vs. Cont1, **** *p* < 0.0001 vs. Cont1, ^##^
*p* < 0.01 vs. PCa1, ^###^
*p* < 0.001 vs. PCa1, ^####^
*p* < 0.0001 vs. PCa1, ^+^ *p* < 0.05 vs. Cont2, ^++++^
*p* < 0.0001 vs. Cont2.

**Table 2 ijms-23-07560-t002:** Incidence of histopathological lesions of the prostate (in number of animals for each type of lesion *per* total number in the group; the coexistence of multiple lesions was observed in some animals).

	Cont1	PCa1	Cont2	PCa2
Non-neoplastic alterations (atypia related to inflammation/functional hyperplasia/metaplasia)	4/8	4/10	5/10	0/14
Pre-neoplastic alterations (dysplasia)	0	3/10	4/10	2/14
Neoplasia	0	3/10	1/10	12/14

**Table 3 ijms-23-07560-t003:** List of proteins obtained by GeLC-MS/MS with differential abundance (*p* < 0.05) in urine between groups (five animals *per* group). Negative or positive fold change values represent the down-regulation or up-regulation, respectively, of the protein. Information about their presence on human prostate glandular cells (hPGC) and detection on human prostate cancer (tissue sections), according to Human Protein Atlas, is also demonstrated. Legend: ^✓^: detected in human prostate cancer (tissue sections); ^✗^: not detected in human prostate cancer (tissue sections); N/D: not detected in hPGC; N/A: not applicable (protein only present in rat).

Gene Name/ UniProt ID	Protein Name	Fold Change				hPGC Protein Expression
		PCa1 vs. Cont1	PCa2 vs. Cont2	Cont2 vs. Cont1	PCa2 vs. PCa1	
Plbd2 ^✓^	Putative phospholipase B-like 2	1.662				medium
Fuca1 ^✓^	Tissue alpha-L-fucosidase 1	1.929				high
Rbp4 ^✓^	Retinol-binding protein 4	1.225		−1.438	−2.352	N/D
Minpp1 ^✓^	Multiple inositol polyphosphate phosphatase 1	1.305				medium
Cdh2 ^✗^	Cadherin-2	−1.428				N/D
Fetub ^✓^	Fetuin-B	−1.294			1.808	N/D
Eno1 ^✓^	Alpha-enolase		1.980		1.625	medium
Ces1c	Carboxylesterase 1C		1.531			N/A
P08932	T-kininogen 2		1.199		1.288	N/A
Q5U2Q3	Ester hydrolase C11orf54 homolog		−1.714		−1.525	N/A
Egf	Epidermal growth factor		−1.375		−1.428	no data
Idh2 ^✓^	Isocitrate dehydrogenase		−1.923	2.071		high
Mep1a ^✓^	Meprin A subunit alpha		−1.337			N/D
Cdh1 ^✓^	Cadherin-1		−1.323	1.215		high
Q6LED0	Histone H3.1			2.370		N/A
Lgals5	Galectin-5			1.411		N/A
Serpina6 ^✓^	Corticosteroid-binding globulin			1.775		N/D
Serping1 ^✗^	Plasma protease C1 inhibitor			1.555		N/D
Clu ^✗^	Clusterin			1.996		N/D
Mme ^✓^	Neprilysin			−3.168		high
Umod ^✗^	Uromodulin			−4.900		N/D
Klk9 ^✗^	Submandibular glandular kallikrein-9			−3.190		N/D
Enpep ^✗^	Glutamyl aminopeptidase			−1.558	−1.549	N/D
Cpq ^✓^	Carboxypeptidase Q			−1.627	−1.616	low
Serpina1 ^✓^	Protein Z-dependent protease inhibitor				1.623	N/D
Igg-2a	Ig gamma-2A chain C region				2.089	N/A
Cp ^✓^	Ceruloplasmin				1.984	N/D
Mug1	Murinoglobulin-1				2.170	N/A
Tf ^✗^	Serotransferrin				1.909	N/D
Lcn2 ^✗^	Neutrophil gelatinase-associated lipocalin				2.202	N/D
Igh-1a	Ig gamma-2B chain C region				2.785	N/A
Hist1h4b ^✓^	Histone H4;Osteogenic growth peptide				1.902	low
Fgg ^✗^	Fibrinogen gamma chain				9.933	N/D
P02761	Major urinary protein				−2.112	N/A
Dpp7 ^✓^	Dipeptidyl peptidase 2				−1.874	medium
Ambp ^✗^	Protein AMBP				−1.492	N/D
Gsn ^✗^	Gelsolin				−1.540	N/D
Hrsp12 ^✗^	Ribonuclease UK114				−1.739	N/D
Psap ^✓^	Sulfated glycoprotein 1				−1.535	High
Mep1b ^✗^	Meprin A subunit beta				−1.681	N/D
P81828	Urinary protein 2				−2.029	N/A
Siae ^✓^	Sialate O-acetylesterase				−1.510	medium
Cspg4 ^✓^	Chondroitin sulfate proteoglycan 4				−1.464	medium
Pvalb ^✗^	Parvalbumin alpha				−1.439	N/D
Cst3 ^✓^	Cystatin-C				−1.361	High
Pigr ^✓^	Polymeric immunoglobulin receptor				−1.684	N/D
Apcs ^✗^	Serum amyloid P-component				−1.339	N/D
Obp1f	Odorant-binding protein				−1.455	N/A
Cd14 ^✗^	Monocyte differentiation antigen CD14				−1.746	N/D

## Data Availability

Data is contained within the article or Appendix A.

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
