# Peer review of "Tracking Prostate Carcinogenesis over Time through Urine Proteome Profiling in an Animal Model: An Exploratory Approach"

_ijms, 2022, doi:10.3390/ijms23147560_

Round 1
Reviewer 1 Report
Moreira-Pais and colleagues have submitted an original article entitled „ Tracking prostate carcinogenesis over time through urine proteome profiling in an animal model: an exploratory approach” for publication in International Journal of Molecular Sciences.
In this study, proteomics analysis of urine samples from healthy animals and animals with prostate cancer was conducted. Changes in the urinary protein level were evaluated at two different time points. Zymography analysis was also conducted to investigate the proteolytic activity in urine. This is a well-written publication reporting on exploratory analysis of urinary peptides in prostate cancer rat model.
The article tackles a very relevant subject of prostate malignancy that places a heavy burden on the society. The article fits in the scope of the journal, but could benefit from additional work.
Comments:
Table 3 is overwhelming, too long and difficult to follow. Simple version of the table, fitting into one page would improve on clarity of the results presentation. To simplify the table, authors can consider for example presenting only most significant findings, removing of up/ down column – its reflected by the fold change, mark proteins detected in human PCa by symbols next to gene name etc.
Table 3 should also provide information about the respective changes in other groups, even when not significant. Lack of significance might be related to the lack of statistical power. Continuity of the changes in different stages of cancer development would be an interesting aspect to investigate.
For those proteins found significant in case- control comparison, it would be interesting to see if their abundance also correlates with scoring of the prostate histological lesions.
Although authors indicate which proteins have been found previously in human, and state that the model reflects well human disease, further investigation at the level of human urinary proteomics data would be helpful to support these claims and as such translational potential/ relevance of the findings.
For that purpose, comparison of the presented data to previously published urinary proteomics human data would be beneficial. Authors could correlate abundance of human orthologs and proteins found in animal models, and check if the proteins found in animal models are correlated with age in patients with prostate cancer, which would further add credibility.
Validation of the most promising findings in human is missing. This could be to some extent compensated by comparison with the published human data.
What are the novel findings brought by the paper? Many of these proteins have been identified before in the context of prostate cancer. Please indicate the novelty aspect.
Page 4, line 154 – Table S1 is cited incorrectly, please correct.
Please add information on the type of the database used for the proteomics search (i.e., reviewed/ reviewed; with isoforms or canonical sequences only).
How proteomics data were normalised.
It is not clear if the p-values presented in Table 3 are adjusted for multiple-testing.
It is advisable to deposit proteomics data in public repositories to support data reuse.
Author Response
We thank the reviewers for their time to assess our manuscript and for their pertinent suggestions and comments. We believe that all the reviewers’ concerns were addressed. In this sense, some modifications were made throughout the manuscript to clarify some points, as well as, in Figure 1C, Figure 1D, Table 3 and Figure S1. New supplementary materials were also submitted - Table S4 and Figure S4 – to attend to the reviewers’ suggestions. Below, the reviewers can find our responses to their suggestions/comments, where the lines and pages numbers correspond to the document with track changes. The changes made along the manuscript are marked using track changes.
Reviewer #1
Moreira-Pais and colleagues have submitted an original article entitled „ Tracking prostate carcinogenesis over time through urine proteome profiling in an animal model: an exploratory approach” for publication in International Journal of Molecular Sciences.
In this study, proteomics analysis of urine samples from healthy animals and animals with prostate cancer was conducted. Changes in the urinary protein level were evaluated at two different time points. Zymography analysis was also conducted to investigate the proteolytic activity in urine. This is a well-written publication reporting on exploratory analysis of urinary peptides in prostate cancer rat model.
The article tackles a very relevant subject of prostate malignancy that places a heavy burden on the society. The article fits in the scope of the journal, but could benefit from additional work.
Comments:
- Table 3 is overwhelming, too long and difficult to follow. Simple version of the table, fitting into one page would improve on clarity of the results presentation. To simplify the table, authors can consider for example presenting only most significant findings, removing of up/ down column – its reflected by the fold change, mark proteins detected in human PCa by symbols next to gene name etc.
R.: We thank the reviewer for the suggestion that was followed. Table 3 was remade to be easier to follow. Some of the information presented in the old version of Table 3 can now be seen in a new supplemental table, the Table S4.
- Table 3 should also provide information about the respective changes in other groups, even when not significant. Lack of significance might be related to the lack of statistical power. Continuity of the changes in different stages of cancer development would be an interesting aspect to investigate.
R.: We thank the reviewer for the suggestion, which was considered. However, to simplify Table 3, we add the information about the proteins that were not significant in the new uploaded table, the Table S4.
- For those proteins found significant in case- control comparison, it would be interesting to see if their abundance also correlates with scoring of the prostate histological lesions.
R.: We thank the reviewer for the pertinent suggestion and correlations between the proteins whose abundance was statistically significant in PCa1 vs. Cont1 and PCa2 vs. Cont2 with the prostate histological score can be observed in a new figure, Figure S4 (page 23). These data were also reported in the “Results” and “Discussion” sections, as follows:
Lines 380-393, page 10: “To better comprehend the role of the proteins significantly expressed in PCa1 vs. Cont1 and PCa2 vs. Cont2 in prostate carcinogenesis, a correlation between their abundance in urine and the prostate histological score was performed (Figure S4). RBP4 did not correlate with the lesion score, as already observed in the heatmap (Figure 2B). ENO1 had a positive correlation between its abundance and the lesion score (r = 0.6374, p<0.05), strengthening the idea that ENO1 is associated with advanced stages of the disease. The content of cadherin-1 had a negative correlation with the lesion score, confirming the MS data (PCa2 vs. Cont2, Table 3). The content of isocitrate dehydrogenase (IDH)2, a key enzyme involved in citrate metabolism, had a negative correlation with the prostate lesion score (r = - 0.6696, p<0.05), suggesting an impairment of oxidative bioenergetics during the progression of prostate carcinogenesis. Interestingly, epidermal growth factor (EGF), which is associated with PCa development [29], had a negative correlation with prostate lesion score (r = - 0.6366, p<0.05), indicating that it is not involved in more advanced stages of the disease.”
Lines 72-74, page 14: “RBP4 levels decreased with aging, probably revealing age-related hormonal changes, which may have contributed to an absence of a significant correlation between RBP4 abundance and prostate lesion score.”
Lines 76-77, page 14: “In fact, ENO1 correlated positively with prostate histological score.”
Lines 93-95, page 15: “In the PCa2 group, lower urinary levels of IDH2, a key enzyme from the Krebs cycle, were detected, and a negative correlation with prostate lesion score was found.”
- Although authors indicate which proteins have been found previously in human, and state that the model reflects well human disease, further investigation at the level of human urinary proteomics data would be helpful to support these claims and as such translational potential/ relevance of the findings.
For that purpose, comparison of the presented data to previously published urinary proteomics human data would be beneficial. Authors could correlate abundance of human orthologs and proteins found in animal models, and check if the proteins found in animal models are correlated with age in patients with prostate cancer, which would further add credibility.
Validation of the most promising findings in human is missing. This could be to some extent compensated by comparison with the published human data.
R.: We thank the reviewer for the suggestions, which were followed. In the new Table S4 can be observed the comparison of the results from our study to the results from published human urinary proteomics data. Along the “Results” and “Discussion” sections, this comparison is also reported, as follows:
Lines 336-338, page 9: “A more detailed information (namely the biological function and cellular component) regarding these 49 proteins and the others whose abundance did not reach statistical significance between groups can be found in Table S4.”
Lines 348-354, page 10: “In order to consolidate these results, we compared them with recent human urinary proteomics data from the literature (Table S4) [24–26]. Regarding PCa1 vs. Cont1, 4 out of 6 proteins were already identified in urine of PCa patients, namely retinol-binding protein 4 (RBP4). Concerning PCa2 vs. Cont2, 4 out of 8 proteins were observed in urine of PCa patients, such as enolase 1 (ENO1). The common proteins between the present study and the clinical studies showed the same tendency regarding its abundance.”
Lines 362-363, page 10: “Similar results were found in humans, with RBP4 being increased in the urine of PCa patients (Table S4).”
Lines 373-376, page 10: “This glycolytic enzyme is expected to have origin in glandular cells from the prostate and increased content was already reported in PCa tissue samples and in the urine of PCa patients (Table S4). Lower urinary levels of cadherin-1 were observed at advanced stages of prostate carcinogenesis (PCa2 vs. Cont2), and similar results were observed in the urine of PCa patients (Table S4);”
Lines 71-72, page 14: “The increase of RBP4 levels was already seem in the urine of patients with PCa [24].”
Lines 77-78, page 14: “Moreover, increased urinary levels of ENO1 were already observed in PCa patients [25].”
- What are the novel findings brought by the paper? Many of these proteins have been identified before in the context of prostate cancer. Please indicate the novelty aspect.
R.: We thank the reviewer for the comment. And yes, in fact many of the proteins have been identified before in the context of prostate cancer. Nonetheless, the evaluation of the interplay of prostate carcinogenesis development with aging is novel in this study and was possible due to the use of an animal model that mimics human prostate adenocarcinoma, allowing to follow prostate carcinogenesis development. Our study also highlights the age effect on prostate remodeling, and suggests protein markers of this process. The novelty aspect of the present study was clarified as follows:
Lines 113-125, page 15: “Overall, urine proteome profiling appears to provide important clues about to the time-course of prostate carcinogenesis. Its proximity to the prostate makes urine the sample of choice to study prostate remodeling, as highlighted herein. Moreover, the use of an animal model allowed the study of the impact of PCa severity and aging on the urinary proteome profile, without being biased by external factors such as lifestyle. Indeed, the data from this study support the analysis of urinary proteins to follow the time-course of prostate carcinogenesis. Elevated levels of RBP4 and cadherin-2 signal early stages of age-related prostate carcinogenesis, while ENO1, T-kininogen 2, and IDH2 represent more advanced stages. Urinary proteome profiling also underscores the aging effect in prostate remodeling, which is characterized by increased levels of Clu and Serpin6. This aging effect should be further investigated in future urinary proteomic studies to identify protein markers of prostate aging that might predict the risk of PCa development.”
- Page 4, line 154 – Table S1 is cited incorrectly, please correct.
R.: We thank the reviewer for the observation, and the information was corrected to Table S4 (now on line 180, page 4).
- Please add information on the type of the database used for the proteomics search (i.e., reviewed/ reviewed; with isoforms or canonical sequences only).
R.: We thank the reviewer for the suggestion, and a clarification was made as follows:
Line 164-166, page 4: “Raw mass spectrometry data were searched against the UniProt Rattus norvegicus protein database (downloaded in February 2018) and only curated proteins and canonical sequences were considered.”
- How proteomics data were normalised.
R.: To fulfill reviewer’s concern and clarify how proteome data was normalized, the subsequent idea was added to the “2. Material and Methods” section, subsection “2.5. LC-MS/MS analysis and protein ID”, as follows:
Lines 171-176, page 4: “Normalization of data between LC-MS runs was performed in MaxQuant based on summation of total peptide ion signals per sample and using a global Levenberg-Marquardt optimization procedure that aimed to minimize the overall changes for all peptides across all samples. Label-free quantification (LFQ) was then performed using MaxQuant based on the determination of the mean pairwise common peptide ratios between samples, requiring at least two peptide ratios per identified protein.”
- It is not clear if the p-values presented in Table 3 are adjusted for multiple-testing.
R.: We thank the reviewer for the observation, which was clarified in the revised version of the manuscript. As stated in Line 292, page 21: “The p-values were adjusted for multiple-testing.”
Nevertheless, the p-values were removed from the novel Table 3 and are now present in Table S4.
- It is advisable to deposit proteomics data in public repositories to support data reuse.
R.: Following reviewer’s advice, the proteomics data are now on the PRIDE repository with the dataset identifier PXD034432. The following sentence was added to the “2. Material and Methods” section, “2.5. LC-MS/MS analysis and protein ID” subsection:
Lines177-179, page 4: “The MS proteomics data have been deposited to the ProteomeXchange Consortium via the Proteomics Identifications (PRIDE) partner repository [17] with the dataset identifier PXD034432.”
Reviewer 2 Report
The manuscript by Moreira-Pais et al describes a rat model of prostate cancer. The main aims and objectives of the research are not entirely clear. These should be clearly stated in the manuscript and also in the abstract. Without these the research reported presents a collection of rather fragmented data which do not make a coherent report.
The most serious issue with the research reported is that only one replicate proteomic analysis seem to have been performed on the set of pooled samples (sections 2.4, 2.5, Results section lines 238-251 and the Tables quoted therein). MS/MS should be replicated. The claims of 321 unique or 219 common proteins identified by MS/MS are not justified without having biological and technical replicates.
Related to the above is another issue is with PC analysis. PC components are not well separated or not separated at all. The ellipses (Fig. 1) look to me rather randomly drawn. The individual data points in Fig.1 could have been the result of the lack of biological and technical replicates, leading to over-interpretation of the data.
The whole of the Results section text should be re-written completely. The current version does not convey the content well. The results are not reported well and are not interpreted.
Other issues:
Text of the manuscript should be re-written with the view to improve quality of English grammar, especially sentence structure. In particular, long sentences should be shortened, e.g. divided into shorter sentences. Do not omit necessary word such as verbs. These issues persist throughout the manuscript. Remove unnecessary enhancing words, e.g. 'tremendously'. Avoid using technical jargon (Materials and Methods section)
Introduction and Discussion sections cite relevant references, but the background is patchy. Some relevant references are not used or not cited. Authors are advised to consider using more up to date urinary proteomics and urinary peptidomic studies, relevant methods, data analysis and reported urine sample collection and processing issues. The latter do affect downstream analysis such as proteomics.
References should be up-to date. For example there is no real need to cite 1970 paper, the technology has improved and changed over the last 50 years. It might be simpler to just describe the gel percentage and the buffers, e.g. using the same format as in the section 2.6. Alternatively, a more up to date suitable reference may be used. Some of the references used to describe the animal models also appear a little out of date (1990s).
The standard deviation used to describe measurement errors is an incorrect measure. Instead standard errors should be calculated and used when referring to and describing means (averages) of experimental measurements.
lines 207-208: the text does not make sense, re-write sentence
lines 257-259: the text does not make sense, re-write
Results section (p.5): arbitrary use of 'p' value cut-off ranging from 0.05 to 0.0001. Possibly in error. Authors should check all values quoted, justify the 'p' cut-off used and be consistent.
Table 1: values should be stated to include significant digits only. The authors can not seriously suggest the animal weight was measured with the accuracy of 1 mg! Also, the correlation between PCa and animal lengths and weights may or may not exist, but there is no causality and without that the data are irrelevant. After all, the rats' lengths (with tails?) is unlikely to become a useful criterion for PCa diagnosis in humans. Authors should remove from the manuscript any data, which do not address the main aims and objectives of the research and which are of no relevance to prostate carcinogenesis, or urine proteomics. Alternatively, if the authors prefer to focus on the rats' biology, the research should be submitted elsewhere.
Table 3 should be redesigned completely and made more compact. Change the table design to avoid having large empty sections. Similar argument applies to Table 1.
Line 243: "... a similar bands profile ..." is far too vague a statement. Authors should provide a quantitative comparison.
The overall manuscript formatting is poor. The authors should pay attention to the layout and take care when positioning figures and tables.
Author Response
Reviewer #2
- The manuscript by Moreira-Pais et al describes a rat model of prostate cancer. The main aims and objectives of the research are not entirely clear. These should be clearly stated in the manuscript and also in the abstract. Without these the research reported presents a collection of rather fragmented data which do not make a coherent report.
R.: We thank the reviewer for the suggestion. The aims and objectives of the study were clarified in the “Abstract” and “Introduction”, as follows:
Lines 31-32, page 1: “The aim of the present study was to add new insights on the progression of prostate carcinogenesis through the analysis of urine proteome.”
Lines 87-91, page 2: “The aim of our study was to gain new insights into the progression of prostate carcinogenesis by urine protein profiling. For that an animal model of chemically- and hormonally-induced adenocarcinoma was used. Two time points of the disease were considered to reflect the progression of prostate lesions over time from pre-neoplastic to neo-plastic alterations.”
- The most serious issue with the research reported is that only one replicate proteomic analysis seem to have been performed on the set of pooled samples (sections 2.4, 2.5, Results section lines 238-251 and the Tables quoted therein). MS/MS should be replicated. The claims of 321 unique or 219 common proteins identified by MS/MS are not justified without having biological and technical replicates.
R.: We understand the reviewer comment, but as stated in subsection 2.4 “SDS -PAGE and in-gel digestion” of the manuscript, 5 different animals per group, i.e. 5 biological replicates, were used to the proteomic analysis, and therefore, no pooled samples were used. The reinforcement of this information was made in the manuscript (line 144, page 3; lines 282, page 7; 284, page 8; and line 289, page 8) and in the captions of Figure 1, Table 3, and Table S1.
- Related to the above is another issue is with PC analysis. PC components are not well separated or not separated at all. The ellipses (Fig. 1) look to me rather randomly drawn. The individual data points in Fig.1 could have been the result of the lack of biological and technical replicates, leading to over-interpretation of the data.
R.: If we understand reviewer comment, all biological replicates (5 per group) were considered in PCA and PLS-DA analysis. As stated in section 2.5 “LC-MS/MS analysis and protein ID” of the revised manuscript the PCA was performed in the MetaboAnalyst v5.0 web portal and the figure produced by MetaboAnalyst cannot be modified. The produced ellipses reflect the proximity or disparity of proteome data within each group. The overlap of ellipses shows, as we highlighted in the manuscript, that urine proteome is similar among groups and only some proteins are different. These proteins were disclosed by volcano plot analysis, which was also performed in the MetaboAnalyst v5.0 web portal. In the revised version of the manuscript, we better explained these analyses in the “Results” section. Regarding the effect of the histological score in urine proteome, in the revised version of the manuscript is presented pairwise PLS-DA to better visualize the effect of malignancy progression on urine proteome.
- The whole of the Results section text should be re-written completely. The current version does not convey the content well. The results are not reported well and are not interpreted.
R.: To fulfill reviewer’s concern, the “Results” section was re-written to better integrate the results and interpret them.
Other issues:
- Text of the manuscript should be re-written with the view to improve quality of English grammar, especially sentence structure. In particular, long sentences should be shortened, e.g. divided into shorter sentences. Do not omit necessary word such as verbs. These issues persist throughout the manuscript. Remove unnecessary enhancing words, e.g. 'tremendously'. Avoid using technical jargon (Materials and Methods section)
R.: To fulfill reviewer’s concern, we thoroughly reviewed the manuscript and grammatical errors, and long sentences were rectified. A proof-reading tool was used to detect language errors and to improve the quality of English grammar.
- Introduction and Discussion sections cite relevant references, but the background is patchy. Some relevant references are not used or not cited. Authors are advised to consider using more up to date urinary proteomics and urinary peptidomic studies, relevant methods, data analysis and reported urine sample collection and processing issues. The latter do affect downstream analysis such as proteomics.
R.: Following the reviewer suggestion and updated/added some references. In the new Table S4 were included studies in human prostate cancer which urine proteome data was deposited in PRIDE, as suggested by reviewer #1. We believe that the comparison of our data with these studies will help in the translation of our findings.
- References should be up-to date. For example there is no real need to cite 1970 paper, the technology has improved and changed over the last 50 years. It might be simpler to just describe the gel percentage and the buffers, e.g. using the same format as in the section 2.6. Alternatively, a more up to date suitable reference may be used. Some of the references used to describe the animal models also appear a little out of date (1990s).
R.: According to reviewer suggestion, the methodology described in subsection 2.4 of the revised version of the manuscript follows the same format as in the section 2.6, as follows:
Lines 143-149, pages 3 and 4: “To a certain volume of urine samples corresponding to 50 µg of total protein (from five animals per group) was add in a proportion of 1:2 (v/v) the sample loading buffer (0.1 M Tris-HCl pH 6.8, 4% (w/v) SDS, 15% (v/v) glycerol, 20% (v/v) β-mercaptoethanol, 0.1% (w/v) bromophenol blue). The urine proteome was then separated in a 12.5% SDS-PAGE gel (37.5:1 acrylamide/bisacrylamide) resolved at 180 V in a Tris-glycine-SDS running buffer (25 mM Tris, 192 mM glycine and 0.1% (w/v) SDS).”
The reference used to describe the animal models was also replaced by a recent one (from 2021).
- The standard deviation used to describe measurement errors is an incorrect measure. Instead standard errors should be calculated and used when referring to and describing means (averages) of experimental measurements.
R.: We understand the reviewer point-of-view. Nonetheless, the aim of the authors is to describe the variation among individuals that contribute to the mean, and thus, the standard deviation (SD) is advised (please see doi: 10.1093/ilar.43.4.244). The authors do not want to describe the precision of the mean, which if it was the case, the standard error of the mean (SEM) should be used or the confidence interval, which is, in fact, preferably than the SEM (according to doi: 10.1093/ilar.43.4.244). Indeed, some journal editors require their authors to use the SD and not the SEM, especially because SEM is a function of the sample size, and so it can be made smaller simply by increasing the sample size (n). The International Journal of Molecular Sciences does not state if SD or SEM should be used, and therefore, the authors believe that SD better reflects the variation among individuals that contribute to the mean. To support our choice:
As stated by Barde et al (doi: 10.4103/2229-3485.100662) “As readers are generally interested in knowing the variability within sample and not proximity of mean to the population mean, data should be precisely summarized with SD and not with SEM.”
As stated by Nagele (doi: 10.1093/bja/aeg087): “The study sample can be described entirely by two parameters: the mean and the standard deviation. The SEM is used in inferential statistics to give an estimate of how the mean of the sample is related to the mean of the underlying population. As the SEM is always smaller than the SD, the unsuspecting reader may think that the variability within the sample is much smaller than it really is. Whereas the SD estimates the variability in the study sample, the SEM estimates the precision and uncertainty of how the study sample represents the underlying population. In other words, the SD tells us the distribution of individual data points around the mean, and the SEM inform us how precise our estimate of the mean is. It is therefore inappropriate and incorrect to present data only as the mean (SEM).”
As stated by Prager et al (doi: 10.1002/cnr2.1150): “Furthermore, unless an author specifically wants to inform the reader about the precision of the study, SD should be reported as it quantifies variability within the sample.”
As stated by Andrade (doi: 10.1177/0253717620933419): The SD is a descriptive statistic that estimates the scatter of values around the sample mean; hence, the SD describes the sample. In contrast, the SEM is an estimate of how close the sample mean is to the population mean; it is an intermediate term in the calculation of the 95% confidence interval around the mean, and (where applicable) statistical significance; the SEM does not describe the sample. Therefore, the mean should always be accompanied by the SD when describing the sample.”
- lines 207-208: the text does not make sense, re-write sentence
R.: Following the reviewer suggestion, the sentence was rewritten as follows (now on lines 239-240, page 5): “Several biological variables were collected at necropsy (Table 1) to evaluate the effect of prostate carcinogenesis on body composition.”
- lines 257-259: the text does not make sense, re-write
R.: Following the reviewer suggestion, the sentence was rewritten as follows (now on lines 306-310, pages 8 and 9): “In order to find quantitative patterns across proteins and experimental groups or prostate histological lesions, a heatmap analysis was performed. Considering the experimental groups, two main clusters (PCa1 and Cont1 vs. PCa2 and Cont2) were highlighted, suggesting a higher effect of aging over PCa on urine protein profile.”
- Results section (p.5): arbitrary use of 'p' value cut-off ranging from 0.05 to 0.0001. Possibly in error. Authors should check all values quoted, justify the 'p' cut-off used and be consistent.
R.: We understand the reviewer for the comment. The significance level (cut-off of p value) was 0.05, as stated in section 2.8 “Statistical analysis”. However, we, in all our articles (as several other authors), report our results as ranges of probabilities, coded with asterisks (or other symbols), in tables and figures, as follows:
p<0.05, for results where 0.01 < p < 0.05
p<0.01, for results where 0.001 < p < 0.01
p<0.001, for results where 0.0001 < p < 0.001
p<0.0001, for results where p<0.0001
- Table 1: values should be stated to include significant digits only. The authors can not seriously suggest the animal weight was measured with the accuracy of 1 mg! Also, the correlation between PCa and animal lengths and weights may or may not exist, but there is no causality and without that the data are irrelevant. After all, the rats' lengths (with tails?) is unlikely to become a useful criterion for PCa diagnosis in humans. Authors should remove from the manuscript any data, which do not address the main aims and objectives of the research and which are of no relevance to prostate carcinogenesis, or urine proteomics. Alternatively, if the authors prefer to focus on the rats' biology, the research should be submitted elsewhere.
R.: We understand the reviewer comment. However, the values present in Table 1 include the digits given by the balance used. In the case of animals, they were weighted in the PLJ 750-3N balance from Kern, which has a readout (d) of 0.001 g. Organs were weighted in a similar balance. If the reviewer think that we should use only 2 decimal numbers we can correct Table 1.
Relative organ’s weight can be calculated by dividing the absolute organ’s weight by body weight or normalized to tibia length, as presented in several studies (doi: 10.1007/s43440-019-00010-3; doi: 10.1371/journal.pone.0080717; doi: 10.1016/j.yjmcc.2020.07.008). In conditions where body weight might change, as is the case of PCa where the loss of body weight, muscle mass and/or fat mass can occur, the assessment of the relative size of the organs can be more accurately quantified by establishing the ratio between organ’s weight and tibia length (doi: 10.1007/s43440-019-00010-3), since the tibia length of rats is an indicator of animal size that is independent of alterations in body composition. In rats, bone growth never ceases (unlike the observed in humans) thus, normalization of absolute organ’s weight to tibia length minimizes the effect of animals’ growth.
The data provided in Table 1 is used to characterize the animal groups and to evaluate the presence of body wasting, which, in the case of cancer, is normally the result of skeletal muscle mass and/or fat mass loss (also known as cancer cachexia). Therefore, the body weight, prostate weight, gastrocnemius muscle weight and adipose tissue weight present in Table 1 are key indicators of animals’ response to prostate carcinogenesis and aging. Furthermore, it consolidates the fact the animal model used mimics what happens in human PCa by resulting in cancer cachexia.
To clarify the importance of anthropometric data in the context of the present study, some information was added to the first paragraph of the “Results” section as follows:
Lines 249-252, page 6: “These results indicate that the increase in prostate weight was not due to an increase in animal size, given by the tibia length. Tibia length is an indicator of animal size. Since bone growth in rats never ceases, normalization of absolute organ’s weight to tibia length is usually used in these animals [22].”
Lines 259-263, page 6: “The decrease of body weight observed in PCa2 animals (vs. Cont2) was due to both a de-crease in skeletal muscle mass (given by the gastrocnemius-to-body weight ratio) and fat mass, which is suggestive of cancer cachexia. These data strengthen the evidence that the animal model herein used recapitulates what occurs in human PCa, since cancer cachexia has a prevalence of 61% in PCa patients [23].”
Nonetheless, if the reviewer #1 and reviewer #2 think that Table 1 does not provide useful information to the present study, the authors will submit it as supplementary material.
- Table 3 should be redesigned completely and made more compact. Change the table design to avoid having large empty sections. Similar argument applies to Table 1.
R.: Following both reviewers’ suggestion, Table 3 was redesigned completely. Some of the information formerly present in Table 3 can now be seen in Table S4. Regarding Table 1, the current configuration (e.g., large empty sections) is derived from the Journal’s layout.
- Line 243: "... a similar bands profile ..." is far too vague a statement. Authors should provide a quantitative comparison.
R.: As suggested by the reviewer, a quantitative comparison was performed. The new graphic is present in Figure S1-B in page 22.
- The overall manuscript formatting is poor. The authors should pay attention to the layout and take care when positioning figures and tables.
R.: We understand the reviewer concern, but the positioning of figures and tables are of the responsibility of the Journal’s layout, since all the figures were submitted in separated tiff. files. Thus, is out of t
Round 2
Reviewer 2 Report
Authors addressed most of the suggestions made by this reviewer. A few points remain to be addressed:
1. The answer to the original point '2' partially addresses the original query. But since no technical replicates were performed (only biological replicates), the precision of the original measurements remains unknown. The authors should clearly declare that limitation of their study in the methods section to avoid potentially misleading the readers.
2. The answer to the original point '12' partially addresses the original query. Nevertheless the version of Table 1 in the revised manuscript requires further corrections.
(1) sample size should be clearly specified for relevant entries in Table 1
(2) values should be rewritten to remove meaningless and misleading expression of the accuracy of measurements. The argument given by the authors about using SD is accepted, but the statement about using accurate balances in not entirely relevant. The measurements errors still exist. The authors have not measured them and can not now accurately estimate them due to the absence of technical replicates. Ignoring the issue of precision of the measurements does not make measurements errors to disappear. The spread of the measurements reported include both the imprecision and the population differences. But SE could still be estimated and such estimate will define the number of significant digits used to report the measurements (weights, lengths and their derivatives) as well as their relevant SD values. SE do not normally include more than two significant digits (authors are referred to statistical texts if further explanation is necessary) and therefore the SDs should not have more than two significant digits quoted either. The expression for the population means should therefore also be re-written. for example, in such a case the Table entry "Body weight (g) 471.728 ± 27.776" should be re-written as "471 ± 28" or even "470 ± 28" The same applies to all Table entries (all measurements and all their derivatives).
3. Table 3 should be reformatted to improve presentation quality.
4. The manuscript should be checked by a native English speaker.
Author Response
List of responses to the reviewer’s comments
We thank the reviewer#2 for his time to assess the revised version of our manuscript and for the suggestions and comments. We believe that his concerns were addressed. In this sense, some modifications were made throughout the manuscript to clarify some points, as well as in Table 1 and respective caption. Below, the reviewer can find our responses to the suggestions/comments, where the lines and pages numbers correspond to the document with track changes. The changes made along the manuscript are marked using track changes.
Reviewer #2
Authors addressed most of the suggestions made by this reviewer. A few points remain to be addressed:
- The answer to the original point '2' partially addresses the original query. But since no technical replicates were performed (only biological replicates), the precision of the original measurements remains unknown. The authors should clearly declare that limitation of their study in the methods section to avoid potentially misleading the readers.
R.: According to the reviewer suggestion, the following sentence was added to lines 228-229 in page 5: “All analyses reported here were performed with biological replicates (no technical replicates were performed).”
- The answer to the original point '12' partially addresses the original query. Nevertheless the version of Table 1 in the revised manuscript requires further corrections.
(1) sample size should be clearly specified for relevant entries in Table 1
(2) values should be rewritten to remove meaningless and misleading expression of the accuracy of measurements. The argument given by the authors about using SD is accepted, but the statement about using accurate balances in not entirely relevant. The measurements errors still exist. The authors have not measured them and can not now accurately estimate them due to the absence of technical replicates. Ignoring the issue of precision of the measurements does not make measurements errors to disappear. The spread of the measurements reported include both the imprecision and the population differences. But SE could still be estimated and such estimate will define the number of significant digits used to report the measurements (weights, lengths and their derivatives) as well as their relevant SD values. SE do not normally include more than two significant digits (authors are referred to statistical texts if further explanation is necessary) and therefore the SDs should not have more than two significant digits quoted either. The expression for the population means should therefore also be re-written. for example, in such a case the Table entry "Body weight (g) 471.728 ± 27.776" should be re-written as "471 ± 28" or even "470 ± 28" The same applies to all Table entries (all measurements and all their derivatives).
R.: According to the reviewer suggestion, the sample size was stated in Table 1 and the data present in this was reformulated. Now the results include 1 significant digit according to similar studies (doi: 10.1002/mus.25221, doi: 10.1371/journal.pone.0024650, doi: 10.1371/journal.pone.0110371).
- Table 3 should be reformatted to improve presentation quality.
R.: If we understand the reviewer comment, we made minor alterations to improve presentation quality of Table 3 without compromise the changes that were made following the recommendation of reviewer#1. Other format issues are managed by the editorial office of the Journal.
- The manuscript should be checked by a native English speaker.
Following reviewer’s suggestion, the language was checked by a native speaker.